# Numerical Rainfall Simulation of Different WRF Parameterization Schemes with Different Spatiotemporal Rainfall Evenness Levels in the Ili Region

**Zhenxia Mu †, Yulin Zhou \*,†, Liang Peng, and Ying He**

College of Water Conservancy and Civil Engineering, Xinjiang Agricultural University,
Urumqi 830052, China; muzhenxia@126.com(Z.M.); pengliang@xjau.edu.cn(L.P.); xjheying@126.com(Y.H.)

\*   Correspondence: zhouyulin19921103@126.com

†   The authors contribute equally to this paper.

**Abstract:** To obtain high-precision precipitation simulation results, different types of rainfall events in the Ili Region are simulated by using the Weather Research and Forecasting (WRF) model with different physical parameterization schemes. According to the spatiotemporal distribution of rainfall evenness, six rainfall events in the Ili Region are divided into four types. Six microphysical parameterization (MP) schemes, five planetary boundary layer (PBL) schemes, and five cumulus (CU) schemes are combined into 14 parameterization members to simulate the rainfall events. It is worth noting that the simulation result sequence of the WRF model (from best to worst) is as follows: type I (events 3 and 5) > type II (events 1 and 6) > type III (event 2) > type IV (event 4). This finding would imply that the WRF model has the best performance for rainfall events with even spatiotemporal distributions, while it is hard to achieve good simulation results for rainfall events with highly uneven spatial and temporal distributions. The results suggest that no single combination of parameterization members provides the best performance for all rainfall events. According to the overall scheme rankings, d, n, and j are the optimal parameterization combination members that accurately describe the spatiotemporal characteristics of the six rainfall events. The study provides guidance for the selection of the physical parameters for the accurate simulation of different types of rainfall events in the arid region of northwestern China.

**Keywords:** WRF; different spatial and temporal evenness; physics parameterization; Ili Region

## 1. Introduction

Precipitation is an important supply of water resources and research on its distribution law is limited due to its own characteristics and the sparseness and inhomogeneity of the current network of observation stations. These limitations represent a bottleneck that restricts the study of hydrological models, especially the simulation of snowmelt runoff in alpine regions [1–3]. Xinjiang is the largest province in China, and it is located in the arid region of northwestern China. Snowmelt runoff in the alpine region is an important source of water for the rivers of Xinjiang. Due to the inconvenient traffic conditions, harsh climatic conditions, and scarce monitoring stations in the alpine region of Xinjiang, research and data on the changes of hydrometeorological factors and the laws of production and confluence are greatly limited.

At present, the Ili Region is the region with the most complete basic data in Xinjiang, and as the basin with the most abundant precipitation in Xinjiang, its precipitation distribution has relatively obvious change characteristics; therefore, we choose the Ili Region as a typical basin in Xinjiang.

Precipitation simulations by the Weather Research and Forecasting (WRF) model are carried out for the Ili Region to explore the temporal and spatial distribution uniformity of precipitation and the model simulation effect. Such simulations can provide a reference for the subsequent analysis of precipitation variation law in the Ili Region and also improve the spatial and temporal resolution and precision of precipitation data. More accurate precipitation data are required to determine the hydrological model parameters, which can improve the accuracy of the runoff simulation [4]. Such data are also of great significance for rationally developing and utilizing water resources in Xinjiang, predicting and early warning for extreme hydrological events, and improving water conservancy projects.

The WRF model is the latest generation of mesoscale dynamic models, and it has achieved good results in meteorological simulations in different research areas around the world. Many studies [5–7] have confirmed that the WRF model can obtain spatiotemporal rainfall simulation results with a high resolution; therefore, the WRF model can effectively improve the runoff simulation accuracy for flood disaster prevention. The WRF model not only provides many physical parameterization schemes but also provides a detailed description of the cloud microphysical process and land dynamic process. Therefore, the WRF model can accurately simulate the complex interactions between different atmospheric processes and can be used to simulate atmospheric processes at various spatial scales ranging from meters to tens of thousands of kilometers and at various time scales that range from hours to decades. Due to the complexity of the rainfall formation and development process, the different physical parameterization schemes in the WRF model will have different impacts on rainfall simulation. Therefore, a large number of sensitivity studies have been conducted on the physical parameterization schemes provided by the WRF model around the world. Daniel Argu Eso [8] found that cumulus (CU) cloud and boundary layer schemes had important effects on precipitation events in Andalusia; Madhulatha and Rajeevan [9] showed that monsoon event simulation in southeast India is very sensitive to the physical scheme selection; Evans et al. [10] reported that optimal physical scheme selection is also specific to specific precipitation scales at specific geographical locations.

With the wide application and development of the WRF model, more physical parameterization schemes are provided by the model, and it is increasingly difficult to determine the optimal parameterization combination schemes for rainfall events. Considering the uncertainty of the different physical parameterization scheme combinations, many numerical rainfall event simulations currently adopt an integrated scheme to determine the optimal physical parameterization scheme combinations, e.g., four different microphysical, two CU and two planetary boundary layer (PBL) parameterization schemes are combined into 16 members to simulate extreme rainfall events in the Ganges basin in India [11]. Two different PBL, two CU, two microphysical parameterization (MP) schemes and three longwave and shortwave radiation (RA) schemes were combined into 36 parameterization combination members to simulate different precipitation processes in South East Australia and examine the sensitivity of each physical parameter [10]. Six microphysical parameterization (MP), six PBL, and two Land Surface Model (LSM) schemes were combined into 36 parameterization members for heavy rainfall prediction in India [12]. Eight parameterization combination members of CU, MP, and PBL schemes were used to evaluate the parameterization sensitivity of the WRF model over Andalusia [8]. These studies show that no single combination of parameterization members has the best performance for all rainfall events.

To determine the performance of the WRF model for different types of rainfall events and different parameterization schemes in the Ili Region, six rainfall events in the Ili Region are divided into four types in this study based on the spatiotemporal distribution of the rainfall evenness. Six MP, five PBL, and five CU schemes are combined into 14 parameterization members to simulate the rainfall events. The layout of this article is as follows: the physical parameterization scheme configurations of the WRF model are provided in Section 2; the WRF model simulation results based on evenness distributions with different spatial and temporal resolutions and the physical sensitivity of the WRF model are examined in Section 3; Section 4 provides a discussion of the simulation results, and a summary of this research is included in Section 4.

## 2. Data and Methodology

### 2.1. The Study Area

The differences in altitude are very large across the Ili Region, and the terrain is complex; the differences in climate are distinct. The Ili Region mainly includes four climatic areas: the valley plain area, the near-mountain valley area, the Zhaosu Basin area, and the high-mountain area. According to statistics of the Kapuqihai hydrology station in this basin, the average annual temperature is 8.8 °C, the extreme maximum temperature is 39 °C, the extreme minimum temperature is –32 °C, the average annual precipitation is 334.02 mm, the average annual evaporation is 1961.04 mm, and the average annual wind speed is 3.3 m/s. Hourly rainfall data were collected from 10 climate stations and 114 telemetric stations (Figure 1).

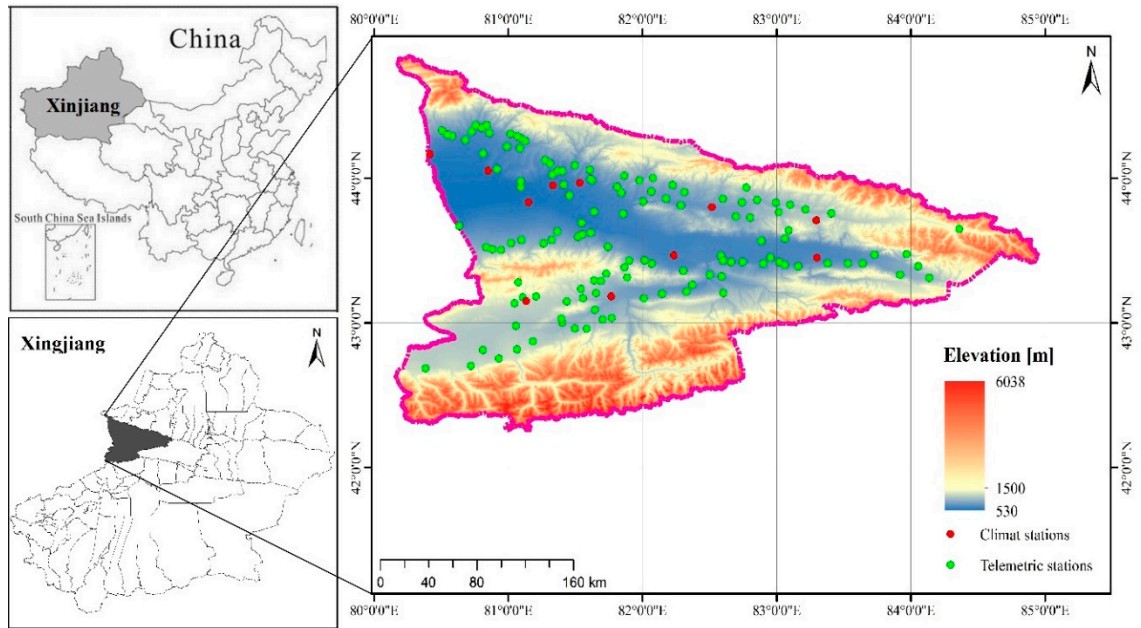

**Figure 1.** The study area.

### 2.2. Model Configuration

The WRF regional climate model (version 3.6) was applied in this study. Model configurations with two nested domains were used, a 9-km resolution parent domain with 300 × 216 grid points, and a 3-km resolution one-way nested domain with 270 × 207 grid points. Only the 3-km resolution domain was analyzed in this study to obtain high-resolution simulation results. The WRF model is initialized from Global Forecast System data (0.5° × 0.5° resolution).

A total of fourteen combination members were produced by the sixteen different physical parameterization schemes. The fourteen parameterization combination members in the WRF model are listed in Table 1 (from scheme "a" to scheme "n"). The sixteen different physical parameterization schemes (include six different MP schemes, five different CU schemes, and five different PBL schemes) are selected to simulate the six events (listed in Table 2). The selected parameterization schemes include six different MP schemes: the WRF Single-Moment 6-class (WSM6) scheme [13], the Thompson (THM) scheme [14], the Purdue Lin (LIN) scheme [15], the Eta-Ferrier (ETA) scheme [16], the Goddard cumulus ensemble (GCE) scheme [17], and the WRF Single-Moment 5-class (WSM5) scheme [18]. Five different CU parameterization schemes are used: the Kain–Fritsch (KF) scheme [19], the Grell–Devenyi (GD) scheme [20] the Grell 3D (G3D) scheme [20], the Betts–Miller–Janjic (BMJ) scheme [21], and the Old Simplified Arakawa-Schubert (OSAS) scheme [22]. The five different PBL schemes are: the Yonsei University (YSU) scheme [23], the Mellor–Yamada–Janjic (MYJ) scheme [24], the Global Forecast System (GFS) scheme [25], the Medium Range Forecast (MRF) scheme [26], and

the Asymmetric Convective Model (ACM2) scheme [27]. The fourteen optimal parameterization combination schemes are summarized in Table 1.

**Table 1.** Fourteen parameterization combination members (member f appears three times).

| Member | MP | | | | | | CU | | | | | PBL | | | | |
|---|---|---|---|---|---|---|---|---|---|---|---|---|---|---|---|---|
| | LIN | THM | ETA | GCE | WSM5 | WSM6 | G3D | GD | KF | BMJ | OSAS | YSU | GFS | ACM2 | MRF | MYJ |
| a | √ | | | | | | | | | | √ | | | | | √ |
| b | | √ | | | | | | | | | √ | | | | | √ |
| c | | | √ | | | | | | | | √ | | | | | √ |
| d | | | | √ | | | | | | | √ | | | | | √ |
| e | | | | | √ | | | | | | √ | | | | | √ |
| f | | | | | | √ | | | | | √ | | | | | √ |
| g | | | | | | √ | √ | | | | | | | | | √ |
| h | | | | | | √ | | √ | | | | | | | | √ |
| i | | | | | | √ | | | √ | | | | | | | √ |
| j | | | | | | √ | | | | √ | | | | | | √ |
| f | | | | | | √ | | | | | √ | | | | | √ |
| k | | | | | | √ | | | | | √ | √ | | | | |
| l | | | | | | √ | | | | | √ | | √ | | | |
| m | | | | | | √ | | | | | √ | | | √ | | |
| n | | | | | | √ | | | | | √ | | | | √ | |
| f | | | | | | √ | | | | | √ | | | | | √ |

The six 24-h events are selected from 2015 and 2016, and these events have different precipitation characteristics, including different spatiotemporal rainfall evenness levels and different 24-h accumulated rainfall amounts. The coefficient of variation Cv is used to evaluate the unevenness level, and the higher Cv is, the more uneven the rainfall event is. According to the spatiotemporal distribution of the rainfall evenness, the six rainfall events in the Ili Region are divided into four types. The duration, accumulated rainfall amount and spatiotemporal Cv of the six rainfall events are listed in Table 2.

To examine the spatial and temporal rainfall evenness across the Ili Region, spatiotemporal Cv values of the rainfall from 1960 to 2016 are calculated. The spatiotemporal distribution of the rainfall in northwest China is much more uneven compared to other regions. Therefore, it is difficult to obtain an even distribution of rainfall events in this area. A threshold value of 5% [28] is selected as the critical value to distinguish even and uneven precipitation events, which was also considered in other statistical analyses in the same area. Through the threshold, we found that the spatial Cv value was 0.794 and the temporal Cv value was 1.343. These findings indicate that rainfall events with a spatial Cv value below 0.794 or a temporal Cv value below 1.343 accounted for 5% of the total rainfall events in the Ili Region from 1960 to 2016.

The rainfall events in type I are characterized by uniform spatial and temporal distributions, with spatial Cv values below 0.794 and temporal Cv values below 1.343. The type II rainfall events exhibit uneven spatial distributions (spatial Cv > 0.794 and temporal Cv < 1.343). The type III rainfall events exhibit uneven temporal distributions (spatial Cv < 0.794 and temporal Cv > 1.343). The type IV rainfall events exhibit uneven spatiotemporal distributions, with spatial Cv values greater than 0.794 and temporal Cv values greater than 1.343. Events 3 and 5 with a spatial/temporal Cv value below 0.794/1.343 are type I rainfall events, and events 1 and 6 with spatial Cv values above 0.794 are type II rainfall events. Event 2 is a type III rainfall event, which has an even temporal rainfall distribution, and event 4 is a type IV rainfall event and has a spatial/temporal Cv value above 0.794/1.343.

**Table 2.** The spatiotemporal distribution and accumulated rainfall of the six events.

| Event ID | Start-End Time (UTC) | Accumulated Rainfall | Spatial Cv | Temporal Cv | Type |
|---|---|---|---|---|---|
| 1 | 27 June 2015 8:00–28 June 2015 8:00 | 26.92 mm | 0.90 | 0.88 | II |
| 2 | 28 June 2015 8:00–29 June 2015 8:00 | 9.57 mm | 1.00 | 1.13 | III |
| 3 | 17 June 2016 8:00–18 June 2016 8:00 | 34.28 mm | 0.69 | 0.73 | I |
| 4 | 18 June 2016 8:00–19 June 2016 8:00 | 10.35 mm | 0.97 | 1.65 | IV |
| 5 | 19 June 2016 8:00–20 June 2016 8:00 | 14.96 mm | 0.51 | 1.30 | I |
| 6 | 8 July2016 8:00–9 July 2016 8:00 | 12.92 mm | 1.11 | 1.29 | II |

*2.3. Evaluation Statistics*

The spatiotemporal Cv (variation coefficient) is calculated to characterize the unevenness level of the spatiotemporal rainfall distribution (in Section 2.2).

$$\text{Cv} = \sqrt{\frac{1}{N}\sum_{i=1}^{N}\left(\frac{x_i}{\bar{x}} - 1\right)^2}, \tag{1}$$

For the spatial distribution, $x_i$ is the 24 h rainfall accumulation at rain gauge $i$, and $\bar{x}$ is the average of $x_i$; $N$ is the number of stations. For the temporal distribution, $x_i$ is the hourly areal rainfall at time $i$, and $\bar{x}$ is the average of $x_i$; $N$ is the number of hours.

The relative error (*RE*) is calculated to verify the WRF simulations of the cumulative rainfall amount (in Section. 3.1.1)

$$RE = \frac{(M-O)}{O} \times 100\%, \tag{2}$$

where $M$ and $O$ are the cumulative rainfall amount of simulation and observation.

The square of the Pearson correlation coefficient ($R^2$) is calculated to verify the WRF simulations of the temporal characteristics of the different rainfall events (in Section 3.1.). The standard deviation ratio ($\sigma$) is calculated to verify the WRF simulations of the temporal characteristics of the different rainfall events (in Section 3.1.2),

$$R^2 = \left(\frac{\sum_{i=1}^{N}(M_i - \bar{M})(O_i - \bar{O})}{\sqrt{\sum_{i=1}^{N}(M_i - \bar{M})^2}\sqrt{\sum_{i=1}^{N}(O_i - \bar{O})^2}}\right)^2, \tag{3}$$

$$\sigma = \frac{\sqrt{\sum_{i=1}^{N}(M_i - \bar{M})^2}}{\sqrt{\sum_{i=1}^{N}(O_i - \bar{O})^2}}. \tag{4}$$

For the spatial dimension, $M_i$ and $O_i$ are the simulation and observation of 24 h rainfall accumulations at each station $i$. $\bar{M}$ is the average of M, $\bar{O}$ is the average of $O$. $N$ is the number of the station, which is 10 climate stations and 114 telemetric stations. For the temporal dimension, $M_i$ and $O_i$ are the average areal rainfall simulation and observation at each time step $i$. $\bar{M}$ is the average of M, $\bar{O}$ is the average of $O$. This time $N$ is 24, which represents the number of the time steps.

The mean absolute error (*MAE*) and the Bias is calculated to evaluate the precipitation process simulation of different parameterization schemes (in Section 3.2.1),

$$MAE = \frac{1}{N}\sum_{i=1}^{N}|M_i - O_i|, \tag{5}$$

$$\text{Bias} = \frac{1}{N}\sum_{i=1}^{N}M_i - O_i. \tag{6}$$

The $M_i$ and $O_i$ are the average rainfall simulation and observation at each time step $i$. $\bar{M}$ is the average of M, $\bar{O}$ is the average of $O$. This time $N$ is 24, which represents the number of the time steps.

To identify the optimal parameterization combination members for the temporal and spatial rainfall simulation results, a total of two different metrics (including *MAE* and $R^2$ scores for the temporal and spatial simulations of the total rainfall, respectively) are calculated and combined into one indicator (the mean metric). Some operations need to be performed on the indicators before ranking the optimal parameterization combination members: the *MAEs* of the fourteen

parameterization combination members are standardized by their respective maxima, and the $R^2$ values of all members are inverted (the temporal and spatial indicators should be calculated separately). After this process, both the *MAE* and $R^2$ values are between 0 and 1, and the closer the value is to 0, the better the simulation results will be. Finally, all the indexes obtained from each scheme were averaged to obtain one indicator for the fourteen parameterization combination members. The mean metric of the fourteen parameterization combination members was compared, temporally and spatially, and the best simulation results were obtained with the minimum comprehensive indicators.

## 3. Results

### 3.1. Performance Evaluation of the WRF Model Simulations

3.1.1. Performance Evaluation of the Temporal Rainfall Simulations

The cumulative curves of the observed and simulated rainfall for the six rainfall events from the 14 members are shown in Figure 2. The simulation results of the cumulative rainfall amount from the 14 members of the physical ensemble are shown in Table 3 and the relative error (*RE*) is calculated between the simulation results of 14 members and the observations (shown in Table 3).

The results show that the performances of the 14 members are quite distinct for different types of rainfall events. In addition, the difference among the 14 members varies greatly for six rainfall events. For example, the difference of *RE*s between member f (75.65%) and member h (−50.47%) is as high as 126.12% for event 2, and the difference of *RE*s between member g (122.22%) and member h (−3.09%) reaches 125.31% for event 4. However, for event 1, the largest difference of *RE* among all the 16 members is only 33.51%. There is great uncertainty in using different combinations of WRF models and physical parameterization to simulate different storm events. It is difficult to tell which parameter combination is the best, but only the combination with the best overall performance can be found.

The shapes of 14 simulated cumulative curves are consistent with the observed ones for events 1, 3, 5 and 6 (type I and type II events), indicating that the simulated rainfall occurrences always keep step with the observations. While for events 2 and 4 (type III and type VI events), the simulated starting and ending times of the rainfall durations are quite different from the observations. It can be determined that type I and type II events have even rainfall distributions in temporal, while the temporal rainfall is unevenly distributed in space for type III and type VI events. It seems that storms with rainfall evenly distributed in temporal tend to have better simulation results in the temporal patterns of rainfall accumulations. Similarly, the simulated results show that the sensitivity of the WRF model parameters is also different for six events. The simulation results of all parameter schemes are slightly different for events 1 and 3, while the differences in the simulation results of the other four events are very large, which means that the parameters of WRF model are sensitive for rainfall events with highly uneven spatial and temporal distributions but are not sensitive for rainfall events with even spatial and temporal distributions.

**Table 3.** The cumulative rainfall (mm) and *RE* of the temporal rainfall simulations.

| Menber | Event 1 Rainfall (mm) | Event 1 RE (%) | Event 2 Rainfall (mm) | Event 2 RE (%) | Event 3 Rainfall (mm) | Event 3 RE (%) | Event 4 Rainfall (mm) | Event 4 RE (%) | Event 5 Rainfall (mm) | Event 5 RE (%) | Event 6 Rainfall (mm) | Event 6 RE (%) |
|---|---|---|---|---|---|---|---|---|---|---|---|---|
| Observed | 26.92 | / | 9.57 | / | 34.28 | / | 10.35 | / | 14.96 | / | 12.8 | / |
| a | 20.72 | −23.03 | 12.27 | 28.21 | 27.41 | −20.04 | 16.49 | 59.32 | 11.04 | −26.20 | 15.74 | 22.97 |
| b | 15.80 | −41.31 | 12.51 | 30.72 | 22.89 | −33.23 | 14.74 | 42.42 | 9.66 | −35.43 | 9.65 | −24.61 |
| c | 18.27 | −32.13 | 12.31 | 28.63 | 25.30 | −26.20 | 17.08 | 65.02 | 11.02 | −26.34 | 11.37 | −11.17 |
| d | 20.23 | −24.85 | 10.38 | 8.46 | 20.10 | −41.37 | 12.91 | 24.73 | 9.18 | −38.64 | 9.83 | −23.20 |
| e | 15.65 | −41.86 | 10.12 | 5.75 | 23.97 | −30.08 | 16.18 | 56.33 | 12.14 | −18.85 | 11.49 | −10.23 |
| f | 18.35 | −31.84 | 16.81 | 75.65 | 24.58 | −28.30 | 15.72 | 51.88 | 11.74 | −21.52 | 11.89 | −7.11 |
| g | 19.79 | −26.49 | 13.10 | 36.89 | 27.14 | −20.83 | 23.00 | 122.2 | 17.83 | 19.18 | 14.45 | 12.89 |
| h | 17.32 | −35.66 | 4.74 | −50.47 | 25.03 | −26.98 | 20.72 | 100.1 | 8.24 | −44.92 | 13.32 | 4.06 |
| i | 18.58 | −30.98 | 9.79 | 2.30 | 24.02 | −29.93 | 18.36 | 77.39 | 15.43 | 3.14 | 11.95 | −6.64 |
| j | 11.70 | −56.54 | 7.95 | −16.93 | 16.04 | −53.21 | 11.61 | 12.17 | 7.04 | −52.94 | 9.23 | −27.89 |
| k | 13.17 | −51.08 | 8.04 | −15.99 | 19.09 | −44.31 | 16.02 | 54.78 | 10.51 | −29.75 | 12.58 | −1.72 |
| l | 15.47 | −42.53 | 12.48 | 30.41 | 20.87 | −39.12 | 14.93 | 44.25 | 13.52 | −9.63 | 14.20 | 10.94 |
| m | 15.17 | −43.65 | 11.70 | 22.26 | 14.20 | −58.58 | 14.56 | 40.68 | 12.97 | −13.30 | 10.54 | −17.66 |
| n | 14.14 | −47.47 | 7.85 | −17.97 | 11.54 | −66.34 | 10.03 | −3.09 | 8.31 | −44.45 | 9.86 | −22.97 |

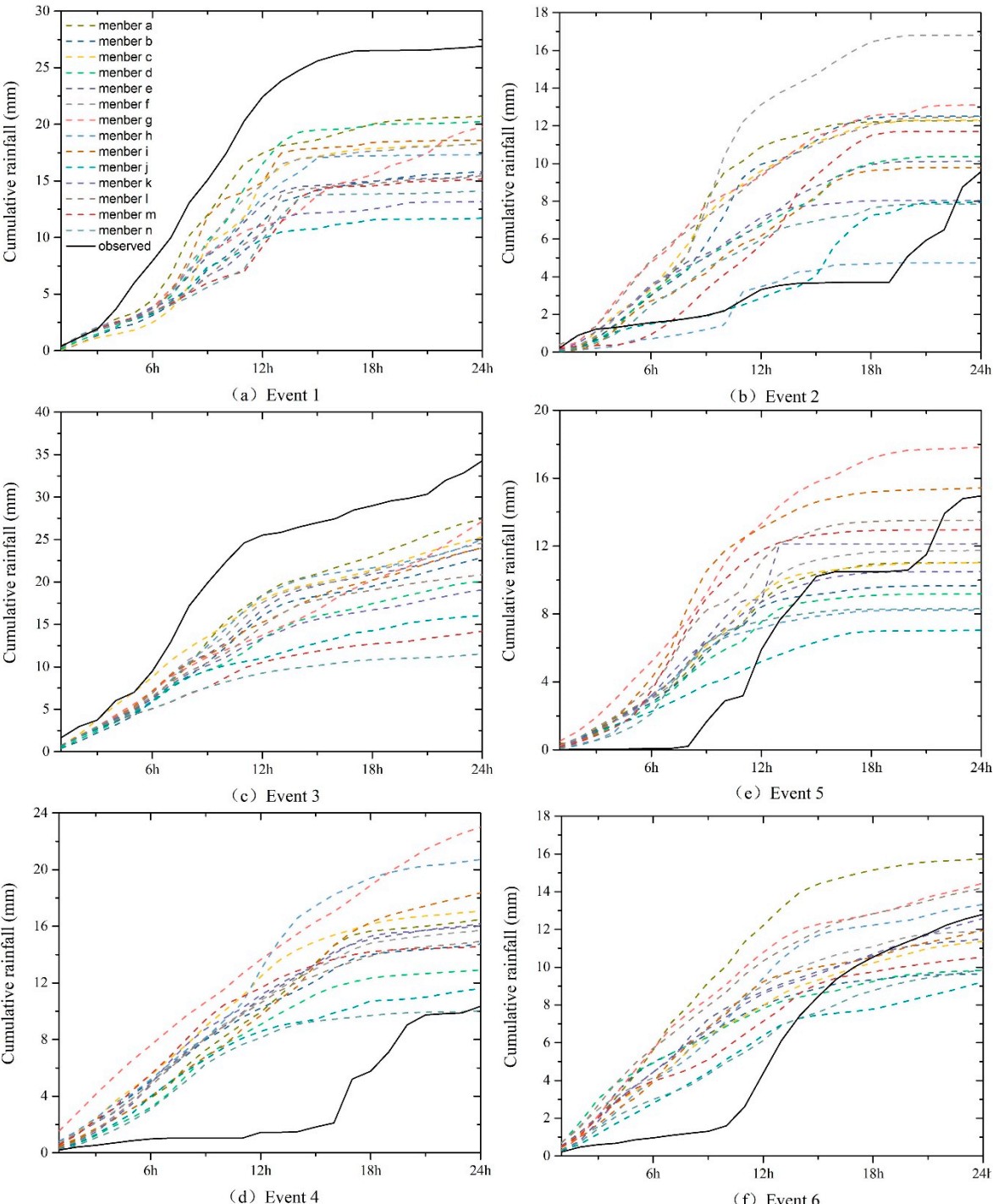

**Figure 2.** The simulation of the 24-h accumulated rainfall amount for the six events.

3.1.2. Performance Evaluation of the Spatial Rainfall Simulations

To compare the simulation results (the average of the simulation results of 14 physical members) of the different storm types in detail, the precipitation center distributions, the maximum, minimum and average of the cumulative rainfall are evaluated for the simulated rainfall distributions, as shown in Figure 3. The Ili Region includes the Yi Ning City and eight counties (Huo Cheng, Yi Ning, Cha Bu Cha Er, Gong Liu, Ni Le Ke, Xin Yuan, Te Ke Si, and Zhao Su, respectively).

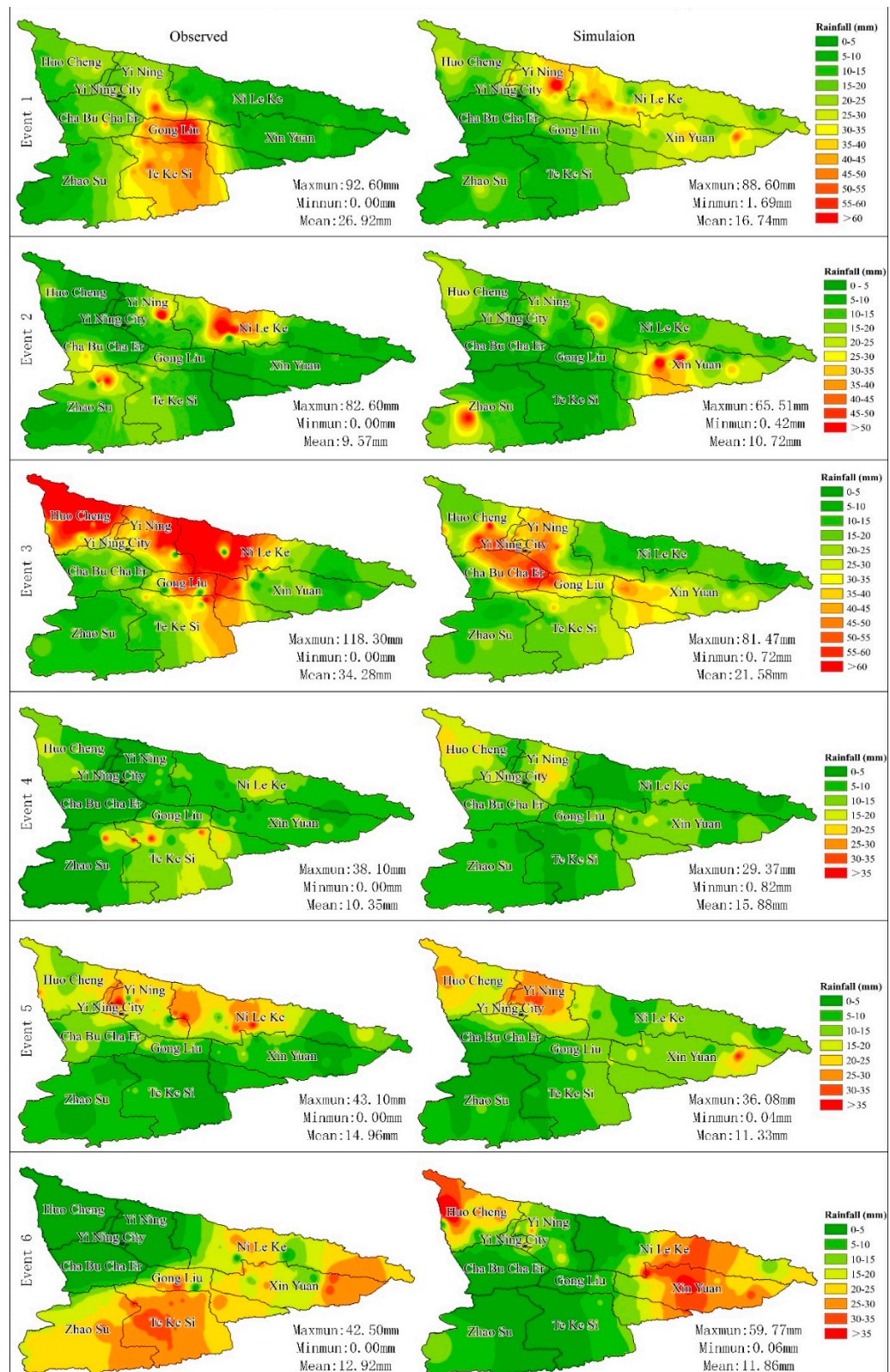

**Figure 3.** The spatial distribution of the rainfall simulation for six events.

The simulations do not accurately capture the distribution characteristics of the maximum precipitation. However, the simulated event 5 captures the maximum rainfall distributions located near the Yi Ning county, which is north of the Ili Region. The precipitation center of event 2 is located at the Ni Le ke, Yi Ning, and Zhao Su counties, and the simulation also results in three precipitation centers at the Xin Yuan, Ni Le ke, and Zhao Su. The northwest corner of the study area and the Ni Le ke county are the maximum precipitation centers of event 3, but the simulated precipitation

distribution only has one center (near to the northwest). Except for event 6, the maximum of the cumulative rainfall of the simulations are all below the observed ones for the other storm events.

It can be seen in Figure 3 that the simulated rainfall has relatively large errors, and the simulation results of the spatial rainfall patterns are always unreliable. The results show that the spatial distribution simulation results of the WRF model of types I and III (events 3, 5, and 2) are much better than those of types II and IV (events 1, 6, and 4). This finding means that precipitation events with even spatial distributions perform better in WRF simulations than those with uneven spatial distributions in simulations of the spatial distribution. In general, the WRF model attains the best performance for rainfall events with even spatial and temporal distributions, while good simulation results are difficult to achieve for rainfall events with highly uneven spatial and temporal distributions.

### 3.2. Impact of Different Parameterization Schemes

#### 3.2.1. Different Parameterization Schemes for the Precipitation Process Simulation

The bias of the hourly rainfall of the 14 physical members for the six events is shown in Figure 4 and the mean absolute error (MAE) is shown in Table 4. The simulations of event 1, as shown in Figure 4a, indicate that the maximum bias range of all members is between −2.46 and 1.97. Most of the 14 parameterization members have the tendency to underestimate the cumulative rainfall, especially from 5–14 h. To evaluate the performance of the parameterization schemes, the mean absolute error of all members was calculated. Members a, d, and f have smaller mean absolute errors than other members, which are 0.40, 0.43, and 0.45, respectively. This result means that the LIN, GCE, and WSM6 physical members have better-simulated results for event 1. Figure 4b shows the performance results of the various parameterization schemes for event 2, reflecting that all members underestimate the hourly rainfall in the last 5 h, and the maximum bias range of all members is between −2.26 mm and 2.74 mm. It is noted that the GD, BMJ, YSU, and MRF physical members are recommended for event 2. Figure 4c shows that all members overestimate the hourly rainfall for most of the time except at 8 h for event 3, and the maximum bias range of all members is between −2.26 mm and 2.74 mm. Members a and h have small mean absolute errors, which are 0.57 and 0.58, respectively. The LIN and GD schemes can be recommended for event 3. The simulation results of event 4 that all members overestimate the hourly rainfall most of the time, as shown in Figure 4d. The maximum bias range of all members is between −2.97 mm and 2.02 mm. The BMJ and MRF schemes can be recommended for event 4, for which the mean absolute errors are 0.63 and 0.65. The simulations of event 5 in Figure 4e show the underestimation of the hourly rainfall from 12–16 h and in the last 3 h, and the maximum bias range of all members is between −2.73 mm and 2.58 mm. The LIN, GCE, and WSM6 schemes are found to perform well. Figure 4f shows the simulation of event 6, with most of the simulated rainfall occurring 10 h earlier for event 6. Most members have the tendency to overestimate precipitation, particularly from 20,160,617 20:00 UTC to 20,160,619 1:00 UTC and from 20,170,708 8:00 UTC to 20,170,708 19:00 UTC. The maximum bias range of all members is between −1.37 mm and 1.33 mm, and the MRF, ACM2, and GD schemes have small mean absolute errors, which are 0.37, 0.39, and 0.42, respectively.

A comparison of the overall simulation errors of all parameter schemes for 6 rainfall events shows the following results. (1) Overall, the simulations of event 1 and event 3 relatively underestimate precipitation, which means that the 14 parameter members have the tendency to underestimate the accumulated precipitation in the event simulation of large cumulative rainfall. The 24-h cumulative precipitation of event 1 was 26.92 mm, and the average cumulative precipitation of the 14 parameter members was 20.70 mm. The 24-h cumulative precipitation of event 3 was 34.28 mm, and the average cumulative precipitation of the 14 parameter members was 27.41 mm. (2) For the other four events (events 2, 4, 5, and 6) with small cumulative rainfall, the beginning of the simulation results shows an overestimated precipitation trend while the latter part of the simulation shows an underestimated precipitation trend.

To determine the optimal parameterization combination members for precipitation process simulation, the simulation results of the different physical schemes are compared in detail. We find that the GD, LIN, and MYJ schemes have the best simulation results in five different CU parameterization schemes, six different MP schemes, and five different PBL schemes.

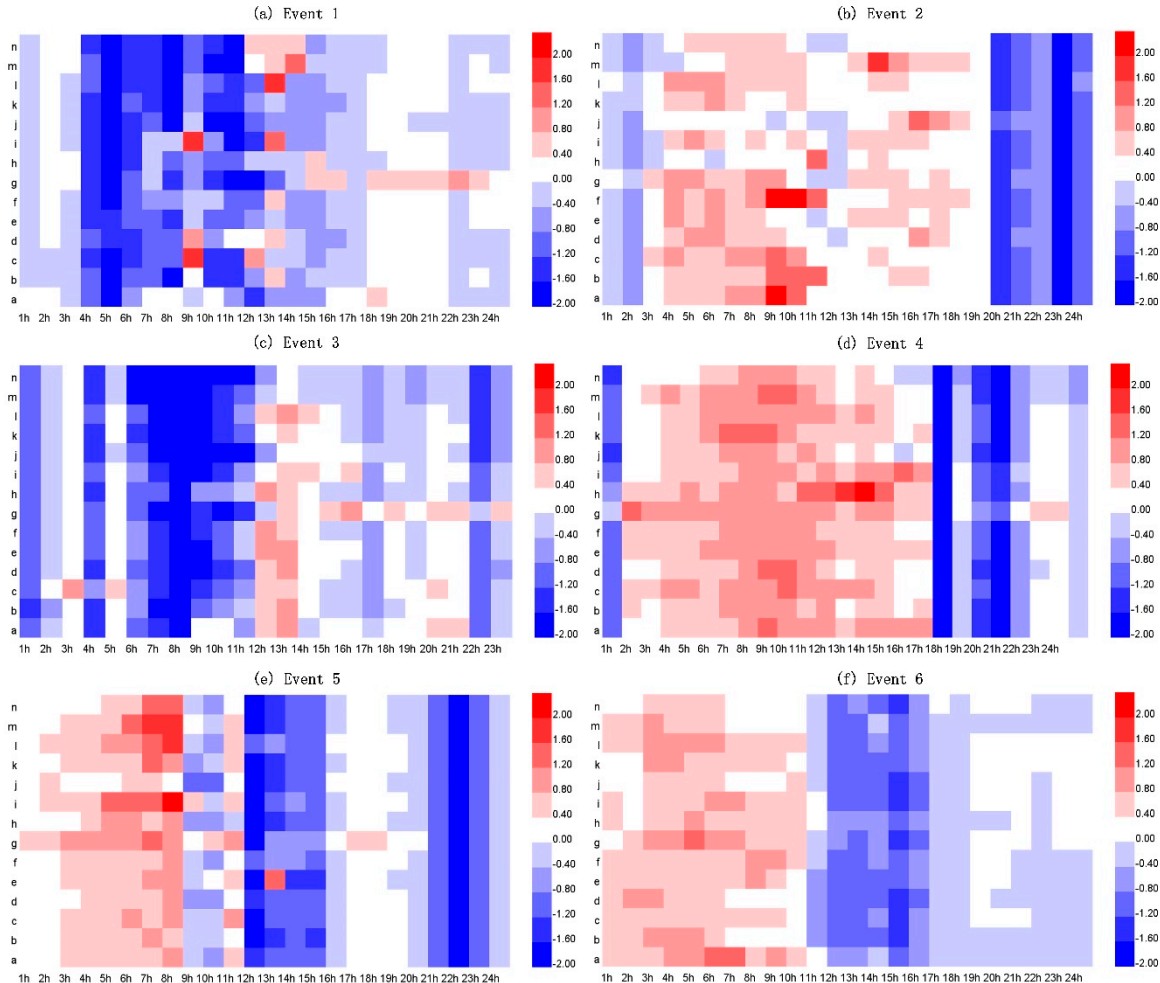

**Figure 4.** The simulation bias of hourly rainfall of 14 physical members for six events.

**Table 4.** The mean absolute error (*MAE*) of the temporal rainfall simulations.

| Event | Member | | | | | | | | | | | | | |
|---|---|---|---|---|---|---|---|---|---|---|---|---|---|---|
| ID | a | b | c | d | e | f | g | h | i | j | k | l | m | n |
| Event 1 | 0.40 | 0.57 | 0.62 | 0.43 | 0.52 | 0.45 | 0.77 | 0.47 | 0.65 | 0.67 | 0.65 | 0.70 | 0.71 | 0.69 |
| Event 2 | 0.65 | 0.67 | 0.64 | 0.57 | 0.56 | 0.82 | 0.64 | 0.40 | 0.57 | 0.48 | 0.46 | 0.61 | 0.65 | 0.48 |
| Event 3 | 0.57 | 0.68 | 0.67 | 0.80 | 0.68 | 0.62 | 0.76 | 0.58 | 0.71 | 0.82 | 0.74 | 0.80 | 0.87 | 0.98 |
| Event 4 | 0.80 | 0.73 | 0.83 | 0.70 | 0.77 | 0.75 | 0.84 | 0.93 | 0.76 | 0.63 | 0.76 | 0.73 | 0.79 | 0.65 |
| Event 5 | 0.61 | 0.63 | 0.67 | 0.62 | 0.72 | 0.59 | 0.77 | 0.70 | 0.81 | 0.63 | 0.70 | 0.68 | 0.76 | 0.69 |
| Event 6 | 0.56 | 0.53 | 0.48 | 0.52 | 0.54 | 0.51 | 0.56 | 0.42 | 0.52 | 0.43 | 0.51 | 0.51 | 0.39 | 0.37 |

3.2.2. Different Parameterization Schemes for the Geographical Distribution Simulation

Taylor diagrams provide a method to display standard deviations and the pattern correlation on one plot. The Taylor diagram (Figure 5) shows the $\sigma$ (calculated by Equation (4) and $R^2$ values between the 14 members and the observations. Figure 5a–c illustrate the standard to evaluate the capability of the different parameterization schemes for the six events. The rainfall simulations with

the five CU parameterization schemes are shown in Figure 5a, the six MP schemes are depicted in Figure 5b, and the five PBL schemes are shown in Figure 5c.

Figure 5a shows that the G3D scheme performs the best among the five CU parameterization schemes for events 1, 2, and 3, and the GD, EX, and KF schemes are the better choices for events 4, 5, and 6, respectively. The BMJ scheme is found to be unsuitable for rainfall simulation at the study sites. Figure 5b indicates that small differences can be observed in the performances of the various parameterization schemes for events 1 and 3, and the WSM6 and LIN schemes are the better choices. The WSM5 scheme performs the best for events 2 and 4, and the THOM and GCE schemes are the better choices for events 5 and 6, respectively. This finding is shown in Figure 5c, where the ACM2 scheme is shown for events 1, 3, and 5, and the MRF scheme performs the best for events 4 and 6. The YSU scheme performed the best for event 1.

Each physical parameterization scheme has different simulation results for the different events, which means that there is no unified optimal physical scheme for the six events. However, there is a clear rule that the simulation effect of the 14 physical parameterization schemes for event 5 is better than those for the other events, which indicates that the characteristics of the precipitation events themselves affect the simulation effect of the model more than the physical parameterization schemes.

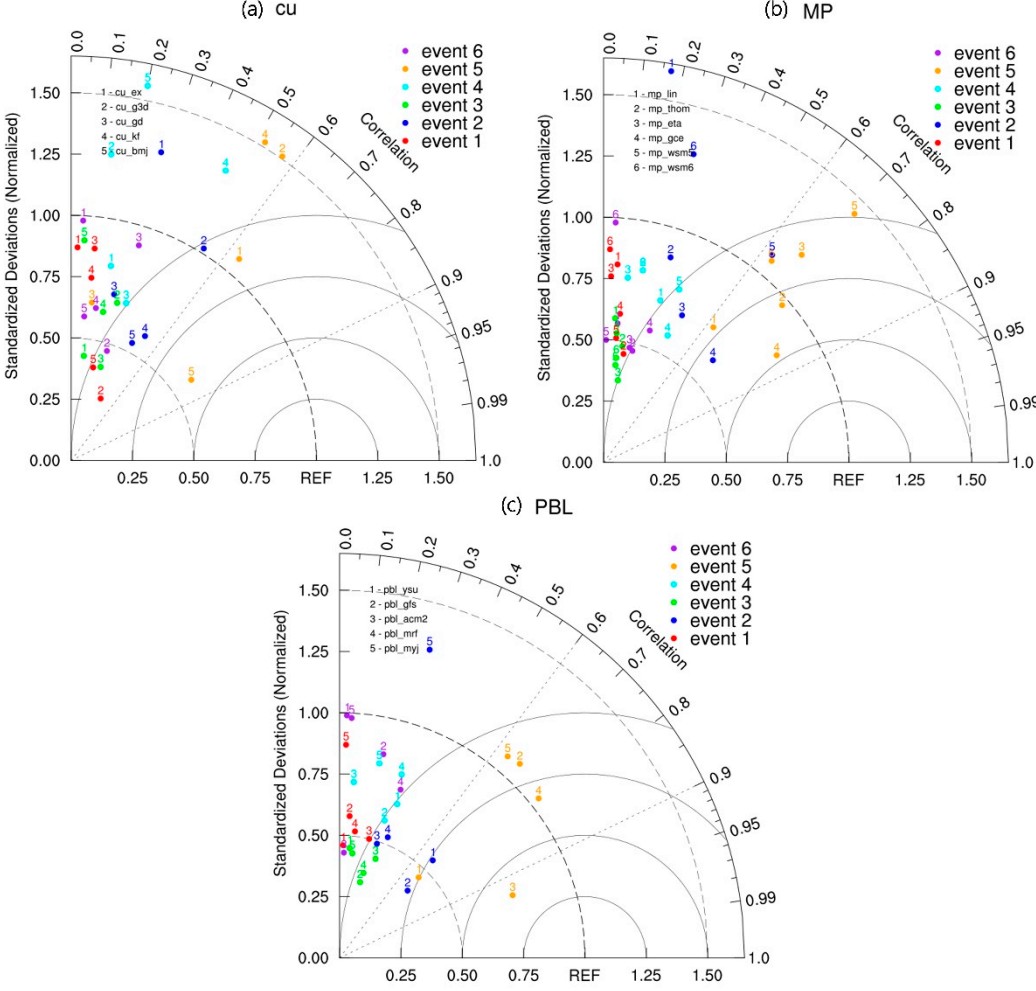

**Figure 5.** Taylor diagrams for the different parameterization schemes for the six events. The colors indicate the different events: event 1 (red), event 2 (blue), event 3 (green), event 4 (cyan), event 5 (orange), and event 6 (purple). The different parameterization schemes have different meanings in the figures.

3.2.3. The Optimal Parameterization Combination Members for Rainfall Simulation

The ranking of the mean metric of the 14 members for six events is shown in Figure 6a, and the ranking for four types is shown in Figure 6b and the mean metric of type I is the average of the mean metric of event 3 and event 5, the mean metric of type II is the average of mean metric of event 1 and event 6, the mean metric of type III is mean metric of event 2, and the mean metric of type IV is mean metric of event 4. At last, the mean metric of all events is shown in Figure 6c. At rank 1 is the member with the best simulation results, which is shown in red. The redder the image, the better the simulation results. However, at rank 14 is the member with the worst performer, which is shown in blue. The darker the blue color, the worse the performance.

As shown in Figure 6a, for event 1 simulations, the ranking of spatiotemporal simulation results are not consistent and member b is the best recommended because it ranks high in the spatiotemporal simulation results. In addition, member j and k are the optimal parameterization combination members for the temporal and spatial rainfall simulation results for event 2 simulations; member h is ranked first in the space-time simulation ranking for event 3 simulations; members k, d, and n are the optimal parameterization combination members for event 4, 5, and 6 simulations, respectively. As shown in Figure 6b, members b and n are recommended for type I and II simulations, respectively; members j and k also are the optimal parameterization combination members for the temporal and spatial rainfall simulation results for the type III simulations; the member j is recommended for type IV. As shown in Figure 6c, the temporal ranking of the 14 physical parameterization members is h > a > j > d > f > n > e > k > c > b > i > l > m > g, from the best to the worst; the spatial ranking of the 14 physical parameterization members is d > n > l > m > j > e > k > b > c > a > i > f > h > g, from the best to the worst. According to the overall scheme rankings, d, n, and j are the best members and consistently and accurately depict the spatial and temporal characteristics of the six event types.

(a) The optimal combination members for Event 1-Event 6

| | Event 1 | | Event 2 | | Event 3 | | Event 4 | | Event 5 | | Event 6 | |
|---|---|---|---|---|---|---|---|---|---|---|---|---|
| | Temporal | Spatial | Temporal | Spatial | Temporal | Spatial | Temporal | Spatial | Temporal | Spatial | Temporal | Spatial |
| a | 1 | 10 | 13 | 13 | 2 | 5 | 8 | 7 | 2 | 8 | 12 | 12 |
| b | 6 | 4 | 12 | 12 | 5 | 9 | 4 | 10 | 5 | 4 | 14 | 9 |
| c | 7 | 12 | 11 | 9 | 4 | 2 | 12 | 9 | 6 | 6 | 5 | 8 |
| d | 2 | 9 | 4 | 2 | 10 | 3 | 5 | 2 | 4 | 2 | 10 | 7 |
| e | 5 | 6 | 7 | 4 | 6 | 7 | 9 | 5 | 8 | 9 | 11 | 4 |
| f | 3 | 14 | 14 | 14 | 3 | 6 | 6 | 8 | 1 | 10 | 7 | 13 |
| g | 14 | 1 | 10 | 7 | 11 | 14 | 3 | 14 | 10 | 13 | 13 | 6 |
| h | 4 | 13 | 2 | 8 | 1 | 1 | 13 | 13 | 11 | 14 | 3 | 11 |
| i | 9 | 11 | 6 | 10 | 7 | 13 | 1 | 12 | 14 | 12 | 9 | 10 |
| j | 10 | 3 | 1 | 5 | 12 | 12 | 2 | 4 | 3 | 5 | 4 | 2 |
| k | 8 | 7 | 5 | 1 | 8 | 10 | 11 | 3 | 9 | 11 | 6 | 14 |
| l | 11 | 8 | 9 | 3 | 9 | 4 | 7 | 6 | 7 | 7 | 8 | 5 |
| m | 13 | 2 | 8 | 11 | 13 | 8 | 14 | 11 | 13 | 1 | 2 | 3 |
| n | 12 | 5 | 3 | 6 | 14 | 11 | 10 | 1 | 12 | 3 | 1 | 1 |

(b) The optimal combination members for type I - type IV

| | type I | | type II | | type III | | type IV | |
|---|---|---|---|---|---|---|---|---|
| | Temporal | Spatial | Temporal | Spatial | Temporal | Spatial | Temporal | Spatial |
| a | 1 | 8 | 4 | 11 | 13 | 13 | 8 | 7 |
| b | 3 | 4 | 11 | 6 | 12 | 12 | 4 | 10 |
| c | 4 | 5 | 8 | 9 | 11 | 9 | 12 | 9 |
| d | 8 | 2 | 5 | 8 | 4 | 2 | 5 | 2 |
| e | 6 | 9 | 9 | 5 | 7 | 4 | 9 | 5 |
| f | 2 | 10 | 2 | 14 | 14 | 14 | 6 | 8 |
| g | 11 | 14 | 14 | 1 | 10 | 7 | 3 | 14 |
| h | 5 | 13 | 1 | 12 | 2 | 8 | 13 | 13 |
| i | 12 | 12 | 12 | 10 | 6 | 10 | 1 | 12 |
| j | 7 | 7 | 6 | 4 | 1 | 5 | 2 | 4 |
| k | 9 | 11 | 10 | 13 | 5 | 1 | 11 | 3 |
| l | 10 | 6 | 13 | 7 | 9 | 3 | 7 | 6 |
| m | 13 | 1 | 7 | 3 | 8 | 11 | 14 | 11 |
| n | 14 | 3 | 3 | 2 | 3 | 6 | 10 | 1 |

(c) The optimal members for all events

| | mean | | |
|---|---|---|---|
| | Temporal | Spatial | |
| a | 2 | 10 | 1 |
| b | 10 | 8 | 2 |
| c | 9 | 9 | 3 |
| d | 4 | 1 | 4 |
| e | 7 | 6 | 5 |
| f | 5 | 12 | 6 |
| g | 14 | 14 | 7 |
| h | 1 | 13 | 8 |
| i | 11 | 11 | 9 |
| j | 3 | 5 | 10 |
| k | 8 | 7 | 11 |
| l | 12 | 3 | 12 |
| m | 13 | 4 | 13 |
| n | 6 | 2 | 14 |

**Figure 6.** The rank of members for all rainfall events based on the mean metric.

*3.3. Impact of Different Spatiotemporal Rainfall Evenness Levels*

To identify the overall WRF model performance in terms of the spatiotemporal results of the rainfall events with different spatiotemporal evenness levels, the errors and correlations of the simulation results are reflected by the mean metric and the temporospatial evenness indicator Cv. It is worth noting that the mean metric and Cv in the temporal dimension have a good linear relationship, and the linear regression coefficient ($R^2$) is 0.73 for the six events (in Figure 7a). This result would imply that the error of the WRF model simulation result increases with increasing spatiotemporal rainfall distribution unevenness. The relationship between the mean metric and Cv in the spatial dimension (shown by Figure 7b) for all events is observed, and the $R^2$ is 0.07. This finding would imply that the lite correlation between spatial simulation effect of the WRF model and spatial rainfall distribution unevenness.

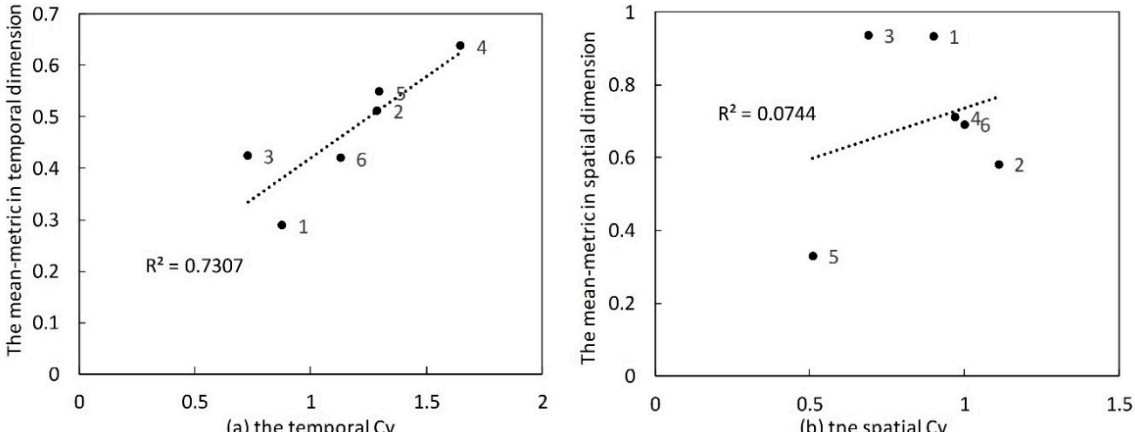

**Figure 7.** The relationship between the mean-metric with the coefficient of variation (Cv).

The performances of the 14 parameterization combination members of the WRF model are characterized by $R^2$ and *MAE* of the temporal and spatial rainfall distribution simulation. The type I rainfall events are characterized by uniform spatial and temporal distributions. The WRF model has the best performance for this rainfall type with even spatial and temporal distributions. Type II rainfall events only have uneven spatial distributions, and type III rainfall events only exhibit uneven temporal distributions, which results in better WRF model simulation results than those for type IV rainfall events, which have uneven spatiotemporal distributions. The WRF model simulation result for the type IV rainfall event is the worst among all rainfall events. The characteristics of the general WRF simulation error will be determined by statistical analysis when the number of rainfall events is increased. It is helpful to establish a modified model, and the accuracy of the WRF model in precipitation simulation will also be improved.

## 4. Conclusions

Different types of representative rainfall events in the Ili Region are simulated using the Weather Research and Forecasting (WRF) model with fourteen different parameterization combination members. According to the spatiotemporal distribution of the rainfall evenness, six rainfall events are divided into four types. It is worth noting that the simulation result sequence of the WRF model (from best to worst) is type I (events 3 and 5) > type II (events 1 and 6) > type III (event 2) > type IV (event 4). The error of the WRF model simulation result increases with increasing spatiotemporal rainfall distribution unevenness. This finding would imply that the WRF model has the best performance for rainfall events with even spatial and temporal distributions, while good simulation results are hard to achieve for rainfall events with highly uneven spatial and temporal distributions.

It is difficult to identify the rainfall type beforehand rainfall forecasting, so it is very important to determine the optimal parameterization combination members for the simulation of all rainfall types in the study area. In this study, six microphysical parameterization (MP) schemes, five planetary boundary layer (PBL) schemes, and five cumulus (CU) schemes are combined into 14 parameterization schemes to simulate the rainfall events. According to the mean metric of the 14 members for the six storm events shown in Figure 6, the mean metric of one certain member for the six storm events is calculated. According to the overall scheme ranking, d, n, and j are the best members, which consistently and accurately depict the spatiotemporal characteristics of the six rainfall types. The characteristics of the general WRF simulation error will be determined by statistical analysis when the number of rainfall events is increased. It is helpful to establish a modified model, and the accuracy of the WRF model in precipitation simulation will also be improved.

This study provides a reference for a comprehensive simulation of different rainfall types in semi-arid and arid regions of China using the WRF models. However, the simulation results of spatial rainfall models are often unreliable, which cannot be directly used for hydrological research. The data assimilation method can effectively improve the simulation accuracy of WRF precipitation simulation models. The following actual telemetry monitoring data in the study area (being deployed) are collected: NCEP (National Centers for Environmental Prediction), ADP (Atmospheric Data Project), Global Upper Air and Surface Weather Observations, and NCEP GDAS (Global Data Assimilation System) Satellite Radiance Data (including ATOVS (Advanced TIROS Operational Vertical Sounder), MHS (Microwave Humidity Sounder) and AMSU (Advanced Microwave Sounding), which are widely used in data assimilation models). The different error calculation methods in the data assimilation model and the errors between the obtained WRF model and the observed data are calculated, and then the simulation results of the precipitation are updated, and the errors are corrected to improve the precipitation simulation accuracy of the WRF model in the study area.

**Author Contributions:** Y.Z. and Z.M. conceived the study, developed the methodology; Y.Z. performed model simulation with the collected data, and analyzed and organized the model results into figures and tables; Z.M. contributed to manuscript revision; supervision, L.P. and Y.H.; project administration, L.P. and Y.H.

**Funding:** This research was supported by the National Natural Science Foundation of China (Grant Nos. 51969029, 51469034 and 51569031), the Natural Science Foundation of the Xinjiang Uygur Autonomous Region (Grant No. 2018D01A16), the key discipline research project of water conservancy engineering of Xinjiang Agricultural University (Grant No. SLXK2018-02).

**Conflicts of Interest:** The authors declare no conflict of interest.

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
