# Peer review of "Numerical Rainfall Simulation of Different WRF Parameterization Schemes with Different Spatiotemporal Rainfall Evenness Levels in the Ili Region"

_water, doi:10.3390/w11122569_

Round 1

Reviewer 1 Report

Point 1: How many parameterization combination schemes have you considered in this work? In the sentence "The physical parameterization settings in the WRF model are listed in Table 1. The selected parameterization schemes include six different MP schemes: the WRF Single-Moment 6-class (WSM6) scheme [12], the Thompson (THM) scheme [13], the Purdue Lin (LIN) scheme [14], the Eta-Ferrier (ETA) scheme [15], the Goddard cumulus ensemble (GCE) scheme [ 16], and the WRF Single-Moment 5-class (WSM5) scheme [17] Five different CU parameterization schemes are used: the Kain – Fritsch (KF) scheme [18], the Grell – Devenyi (GD) scheme [19 ] the Grell 3D (G3D) scheme [20], the Betts – Miller – Janjic (BMJ) scheme [21], and the Old Simplified Arakawa-Schubert (OSAS) scheme [22] The five different PBL schemes are: the Yonsei University (YSU) scheme [23], the Mellor–Yamada – Janjic (MYJ) scheme [24], the Global Forecast System (GFS) scheme [25], the Medium Range Forecast (MRF) scheme [26], and the Asymmetric Convective Model (ACM2) scheme [27]. The fourteen optimal parameterization combination schemes are summarized in Table 1." you have said that the parameterization combination schemes are fourteen when in reality counting them are sixteen. Also in table 1 you can see sixteen parameterization combination schemes.

Figure 4 shows the results of fourteen parameterization combination schemes.

Figure 5 shows the results of sixteen parameterization combination schemes.

Figure 6 shows the results of fourteen parameterization combination schemes.

Response 1: A total of fourteen combination members were produced by the sixteen different physical parameterization schemes. The fourteen parameterization combination members have considered in this work from scheme a to scheme n are listed in Table 1, but the members f appears three times (the member f is highlighted in red), that is why in reality counting them are sixteen. And in Figure 5 the member f also appears three times.

We make necessary corrections in the revised draft, and the corrections will be shown in red font in the revised draft. The corrections as follows: A total of fourteen combination members were produced by the sixteen different physical parameterization schemes. The fourteen parameterization combination members in the WRF model are listed in Table 1 (from scheme a to scheme n). The sixteen different physical parameterization schemes (include six different MP schemes, five different CU schemes, and five different PBL schemes) are selected to simulate the six events.

OK.

Point 2: In the sentence "To identify the optimal parameterization combination schemes for the temporal and spatial rainfall simulation results, a total of four different metrics (including MRE and R scores for the temporal and spatial simulations of the total rainfall, respectively) indicator (the mean metric)." it is not clear which metrics are used to evaluate the indicator (the mean metric). Are they perhaps: σ, R, R2 and MRE?

Response 2: We revise and recalculate the index, a total of two different metrics used to evaluate the indicator (the mean metric). We make necessary corrections in Sect.3.2.3 as follows: To identify the optimal parameterization combination members for the temporal and spatial rainfall simulation results, a total of two different metrics (including MAE and R2 scores for the temporal and spatial simulations of the total rainfall, respectively) are calculated and combined into one indicator (the mean metric).

OK.

Point 3: In the sentence “It is worth noting that the mean metric and Cv in the temporal dimension have a

good linear relationship, and the linear regression coefficient (R2) is 0.73 for the six events (in Figure 7 (a)). The linear regression coefficient (R2) is 0.91 for all events except for events 1 and 3.” why use R for both the Pearson correlation coefficient and for the linear regression coefficient?

Response 3: We used two correlation coefficients in this study: R (Pearson correlation coefficient) and R2 (the linear regression coefficient). In our study, the Pearson correlation coefficient calculated is the square root of the linear regression coefficient. This means that the values calculated by these two indicators represent the same meaning, and both represent the correlation between them. We unified the correlation coefficient of the whole text and decided to use R2. The equation 4 was deleted in the revised draft and the R in Sect.3.1.2 was modified. The modified content has been marked in red in the paper.

OK.

Point 4: Six events are enough to fine-tune the technique proposed in this work, but they are few to exhaustively study the behavior of the different parameterization combination schemes. I hope that in the next work this technique can be applied to a greater number of case studies.

Response 4: In the future, we will continue to improve our basic data in the next work this technique can be applied to a greater number of case studies.

OK.

Minor comments:1) In the sentence “Two different PBL, two CU, two MP and three RA schemes …”, the meaning of RA is not specified. 2) In equation 1 n must be replaced with N. 3) In Figure 2, in Event1 and Event 2 is not possible to distinguish the black line from the gray one.

Response: Thank you for your advice. 1) We have specified the meaning of RA in “Two different PBL, two CU, two MP and three longwave and shortwave radiation (RA) schemes …”. 2) We have corrected the equations (1), and changed N to represent the n in the equations. 3) We have corrected the Figure 2.

OK.

Author Response

Thank you for your valuable comments on this article.

Reviewer 2 Report

Basically the paper is interesting and fits into the scope of the journal, however, I have some questions, notes, and suggestions in the attached text.

Author Response

    Thank you for your valuable comments on this article. We have answered and revised all the questions raised by the reviewer 2. The specific reply is in " Response letter-2.pdf", and the specific revised content has been marked red in the revised draft.

Reviewer 3 Report

Dear authors,

The title of the manuscript by Mu et al. seems to be very useful and practical research. The structure of the manuscript is logically sound and the effort of authors is impressive as well. However, the paper has a very little scientific contribution. There is no practical approach to this research in the future. At this moment, the reviewer rejects this manuscript for publication. Please refer to the enclosed attachment.

Anonymous

Reviewer

Author Response

    Thank you for your valuable comments on this article. We have answered and revised all the questions raised by the reviewer 2. The specific reply is in " Response letter-3.pdf", and the specific revised content has been marked red in the revised draft.

Round 2

Reviewer 2 Report

The paper is improved significantly. Most of my comments were addressed. However, I am still not convinced, that fitting a function on four points out of six makes sense.

The format of the references is still not eligible.

Author Response

Thank you for your valuable comments on this article. According to your suggestions, we make necessary corrections and responses in 'Response letter-2.pdf'.

Reviewer 3 Report

Reviewing the revised version of the manuscript, the reviewer appreciates the persistence of the authors. The authors tried to incorporate most of the comments. However< the reviewer doesn't satisfy with some of the responses to critical comments that posed in the previous version. for example,
The response of point 1 is not satisfactory. The reviewer doesn't see the important aspect of considering CV. The current response is too general and not specific to the choice of CV.
Even though the authors wisely responded the points 6 and 7 with the addition of Table 3 and elaboration of Figure 2, in most of the events the simulated ones are not identical to observed ones. The reviewer thinks that concentrating on R2 value is not a good idea to explain the goodness of the parameterization and relevant combination schemes. It is ought to say that such schemes depend on proper optimization schemes as well. In this regard, the reviewer recommends comparing other statistics of simulated patterns with those of observed ones. Such statistics may include a close approximation of histogram, the geometry of the events and beyond.
What kind of data assimilation studies are recommended to improve the simulation results of the WRF model? Such issues need special attention. The authors' thoughts are not implicitly clear.
Having some of the critical issues, the reviewer is not still convinced of the result. Moreover, the novelty of the research is not properly stated. Still the paper reflects just the application of data in the existing model. For this reason, the reviewer apologizes to say that this paper still needs major revision.
Good luck for the authors to make this article more sound and scientifically better.

Author Response

       Thank you for your valuable comments on this article. According to your suggestions, we make necessary corrections and responses 'Response letter-3.pdf'.

Round 3

Reviewer 3 Report

Most of the comments were incorporated. However, there are some specific errors as mentioned below:

In the caption of Figure 7, "tne"  should be "the" In line 403, there is an extra close parenthesis, ")". The new sentence in lines 398 - 406 is too long. There are also missing "comma"s in the sentence while listing "NCEP, ...."

Once, these comments were addressed and spellings were checked, I am okay with this manuscript.

Congratulations to the authors for having new publications.

Best regards,

Anonymous reviewer.

This manuscript is a resubmission of an earlier submission. The following is a list of the peer review reports and author responses from that submission.

Round 1

Reviewer 1 Report

Major comments

1) How many parameterization combination schemes have you considered in this work?

In the sentence "The physical parameterization settings in the WRF model are listed in Table 1. The selected parameterization schemes include six different MP schemes: the WRF Single-Moment 6-class (WSM6) scheme [12], the Thompson (THM) scheme [13], the Purdue Lin (LIN) scheme [14], the Eta-Ferrier (ETA) scheme [15], the Goddard cumulus ensemble (GCE) scheme [ 16], and the WRF Single-Moment 5-class (WSM5) scheme [17] Five different CU parameterization schemes are used: the Kain – Fritsch (KF) scheme [18], the Grell – Devenyi (GD) scheme [19 ] the Grell 3D (G3D) scheme [20], the Betts – Miller – Janjic (BMJ) scheme [21], and the Old Simplified Arakawa-Schubert (OSAS) scheme [22] The five different PBL schemes are: the Yonsei University (YSU) scheme [23], the Mellor–Yamada – Janjic (MYJ) scheme [24], the Global Forecast System (GFS) scheme [25], the Medium Range Forecast (MRF) scheme [26], and the Asymmetric Convective Model (ACM2) scheme [27]. The fourteen optimal parameterization combination schemes are summarized in Table 1." you have said that the parameterization combination schemes are fourteen when in reality counting them are sixteen. Also in table 1 you can see sixteen parameterization combination schemes.

Figure 4 shows the results of fourteen parameterization combination schemes.

Figure 5 shows the results of sixteen parameterization combination schemes.

Figure 6 shows the results of fourteen parameterization combination schemes.

2) In the sentence "To identify the optimal parameterization combination schemes for the temporal and spatial rainfall simulation results, a total of four different metrics (including MRE and R scores for the temporal and spatial simulations of the total rainfall, respectively) indicator (the mean metric)." it is not clear which metrics are used to evaluate the indicator (the mean metric). Are they perhaps: σ, R, R2 and MRE?

3) In the sentence “It is worth noting that the mean metric and Cv in the temporal dimension have a good linear relationship, and the linear regression coefficient (R2) is 0.73 for the six events (in Figure 7 (a)). The linear regression coefficient (R2) is 0.91 for all events except for events 1 and 3.” why use R for both the Pearson correlation coefficient and for the linear regression coefficient?

4) Six events are enough to fine-tune the technique proposed in this work, but they are few to exhaustively study the behavior of the different parameterization combination schemes. I hope that in the next work this technique can be applied to a greater number of case studies.

Minor comments

1) In the sentence “Two different PBL, two CU, two MP and three RA schemes …”, the meaning of RA is not specified.

2) In equation 1 n must be replaced with N.

3) In Figure 2, in Event1 and Event 2 is not possible to distinguish the black line from the gray one

Reviewer 2 Report

Review of “ Numerical rainfall simulation of different WRF parameterization schemes with different spatiotemporal rainfall evenness levels in the Ili Region”, by Y. Zhou and co-authors (water-587243)

In this manuscript, Zhou and colleagues analyse the ability of WRF in simulating different precipitation events, using a variety of combinations of microphysical (MF), Cumulus (Cu), and planetary boundary-layer (PBL) parameterization schemes. The objective was to determine what would be the best combination of parameterization schemes that would efficiently simulate (or predict) precipitation in the region.

They concluded that all combinations work similarly well in cases of uniform rain in time and space, but simulations results were not adequate for situations where rain was unevenly distributed instinctively of the parameterization combination.

The premise of the work is interesting, but flawed. It relies on the assumption that the parameterization schemes, and them only, may determine the outcome of the simulation. What about initial and boundary conditions? Do the simulations assimilate the observations? Six “events” were listed, but 1 & 2 are continuous, as are 3-5. Only event 7 is a “stand alone” event. Are simulations improved if run continuously? Where they run continuously, or were they reinitialised for each run?

Regarding the comparisons between model outputs and rain gauge data, it is not clear what data did the authors used. Figure 1 shows two types of stations: climate and telemetric stations. What does each measure? For the comparisons,an average of all gridpoints (within the study area domain) were used for WRF results, or did you perform an average of all gridpoints closest to each station? Since the simulation domain include the mountainous area around the valley, could this choice affect the results? Do error distribution have a spatial signature? I.e., are they larger in the slopes or high areas?

That regional numerical models have difficulty simulating precipitation in cases of uneven spatial and temporal distribution is hardly a new result. What is missing here is to investigate the underlying factors for that (for example, was there a low-pressure surface system that the model did not correctly simulate? Was the orographic effect misrepresented?  

Overall, English is good, but style must be improved. The Introduction list a number of studies that addressed similar questions to those studied here with different areas of study, but is not engaging enough. Similarly, the interest in studying the Ili Region must be highlighted. Why was this region chosen? 

Figure 3 is nearly illegible. Also, It would be interesting to get one single color scale for all events in Figure 4 - that would allow a visual comparison between different events as well.